# The Emergence of Japanese Encephalitis in Australia and the Implications for a Vaccination Strategy

**DOI:** 10.3390/tropicalmed7060085

**Published:** 2022-05-29

**Authors:** Luis Furuya-Kanamori, Narayan Gyawali, Deborah J. Mills, Leon E. Hugo, Gregor J. Devine, Colleen L. Lau

**Affiliations:** 1UQ Centre for Clinical Research, Faculty of Medicine, The University of Queensland, Herston 4029, Australia; 2Mosquito Control Laboratory, QIMR Berghofer Medical Research Institute, Herston 4006, Australia; narayan.gyawali@qimrberghofer.edu.au (N.G.); leon.hugo@qimrberghofer.edu.au (L.E.H.); greg.devine@qimrberghofer.edu.au (G.J.D.); 3Dr Deb The Travel Doctor, Travel Medicine Alliance, Brisbane 4000, Australia; email@drdeb.com.au; 4School of Public Health, Faculty of Medicine, The University of Queensland, Herston 4006, Australia

**Keywords:** *Culex*, emergence, intradermal, mosquito, travel, vaccine

## Abstract

Japanese encephalitis (JE) is the leading cause of viral encephalitis in Asia. Until 2022, only six locally transmitted human JE cases had been reported in Australia; five in northern Queensland and one in the Northern Territory. Thus, JE was mainly considered to be a disease of travellers. On 4 March 2022, JE was declared a ‘Communicable Disease Incident of National Significance’ when a locally acquired human case was confirmed in southern Queensland. By 11 May 2022, 41 human JE cases had been notified in four states in Australia, in areas where JE has never been detected before. From this perspective, we discuss the potential reasons for the recent emergence of the JE virus in Australia in areas where JE has never been previously reported as well as the implications of and options for mass immunisation programs if the outbreak escalates in a JE virus-immunologically naïve population.

## 1. Introduction

Japanese encephalitis (JE) is the leading cause of viral encephalitis in Asia [1], with an estimated 100,000 cases and 25,000 deaths per year [2]. Until 2022, the only known locally transmitted human cases of JE in Australia occurred in 1995 on Badu Island in the Torres Strait (three human cases, two deaths) [3] and in 1998 on Badu Island (one case, fully recovered) and the Cape York peninsula (one case, recovered with long-term cognitive challenges), all in northern Queensland [4]. An additional death, associated with exposure in the Tiwi Islands in the Northern Territory occurred in early 2021 [5].

JE became a nationally notifiable disease in Australia in 2001. Until 2016, only 12 cases in returned travellers had been reported to the National Notifiable Diseases Surveillance System [6]. The latest recorded case was in a traveller returning from Bali, Indonesia, in 2018, who died from the infection [7]. In line with the epidemiological risk profile, the current Australian Immunisation Handbook recommendations for JE vaccination are targeted to: (i) people who live or work on the outer islands in the Torres Strait; (ii) travellers spending one month or more in endemic areas during transmission seasons; and (iii) laboratory workers exposed to the virus [8].

## 2. Emergence of Japanese Encephalitis Virus

On 25 February 2022, the presence of the JE virus (JEV) was confirmed in samples from a commercial pig farm in Queensland [9]. Notifications from other piggeries in South Australia, southern and western New South Wales and northern Victoria quickly followed [10]. On 4 March 2022, JE was declared a ‘Communicable Disease Incident of National Significance’ when a locally acquired human case was confirmed in southern Queensland. By 11 May 2022, 41 (28 confirmed and 13 probable) human JE cases had been notified in Australia (NSW (*n* = 13), Victoria (*n* = 14), South Australia (*n* = 9) and Queensland (*n* = 5)) [11] (Figure 1). It has been estimated that < 1% of adult JE infections are symptomatic [12]; thus, thousands of human infections are likely to have occurred.

The wide geographic distribution of recent cases in four states and territories in Australia within a short span of time, particularly in areas where JE has never been detected before, is of great concern. The reasons for the recent emergence of the JEV in Australia are unclear. Molecular genotyping identified genotype IV during the current Australian outbreak [13], but also in Papua New Guinea (PNG) [9], Indonesia [14] and the Tiwi Islands in 2021 [15], implicating a long-range virus dispersal by migrating wading birds, the primary enzootic hosts of the virus, from PNG and Asia [16].

The opportunistic dispersal of migratory birds to inland regions of Australia may have been driven by several months of heavy rainfall associated with the occurrence of a La Niña weather system, leading to the formation of temporary inland wetlands. These conditions encourage the proliferation of mosquitoes [17], including *Culex annulirostris*, which has been historically considered to be the major vector of the JEV in Australia [18]. Other potential vectors are also present [18,19].

Where viraemic birds and high mosquito densities have coincided near piggeries, the conditions may have facilitated transmission to pigs and then to nearby humans. Domestic pigs are associated with most human cases of JE globally [15] and are a major amplifying host [16]. Most other mammals, including humans, do not amplify the virus to the degree needed to infect mosquitoes and facilitate onward transmission [15]. It is possible that in Australia, the transmission of JE has also been partly facilitated by the presence of a large, widely distributed feral pig population [20] as well as the recent establishment of other highly competent mosquito vectors, *Culex gelidus* [21] and *Culex tritaeniorhynchus* [22], in northern Australia. Their current distributions are poorly defined.

## 3. Japanese Encephalitis Vaccines

There is no specific treatment for JE, but effective vaccines are available for humans. There are four vaccine classes currently available worldwide (i.e., inactivated mouse brain-derived, inactivated Vero cell-derived, live recombinant and live attenuated). Two vaccines are licensed in Australia: Imojev (live recombinant) and JEspect (inactivated Vero cell-derived). Both are highly immunogenic [23] and safe [24]. Imojev has the added advantage that it only requires one dose and no boosters are needed in adults whereas JEspect requires two primary doses (28 days apart) and booster doses are recommended 1–2 years after the primary vaccination if there is an ongoing risk of infection [8]. Imojev is a live vaccine and is thus contraindicated in immunocompromised individuals and pregnant women. Receiving other live vaccines a month before or after receiving Imojev is not recommended. In contrast, JEspect is an inactivated vaccine and can be utilised in people who are immunocompromised, pregnant or breastfeeding [8].

For mass immunisation programs, the one-dose schedule of Imojev without the need for boosters in adults (and a booster dose 1–2 years later in children who are at an ongoing JE risk) is logistically easier to implement. Although a single dose of Imojev provides long-term immunity in individuals living in endemic areas [25], the long-term immunogenicity of Imojev has not been examined in non-endemic areas where the population does not have a repeated exposure to the JEV. In endemic countries, the large majority of JE cases—including severe cases—have been reported in children and young adults [26]. Thus, in many Asian countries (e.g., China, Malaysia and Vietnam), a JE vaccination is included in the childhood immunisation program [27]. The incidence of severe JE cases in the adult population is lower due to combined factors of childhood immunisation and a repeated natural exposure.

However, in situations where the virus is introduced to JEV-naïve populations (e.g., Australians), it is likely that severe JE cases will affect all age groups. The Australian government has purchased 130,000 JE vaccine doses, but many more may be needed to protect the population at risk if JE-infected mosquitos are detected near large population centres or if the virus persists through the 2022 winter and causes larger outbreaks next summer [28]. It is notable that the major Australian JEV vector, *Culex annulirostris*, can disperse several kilometres per day [29]. The recommendations for priority vaccinations have expanded to those with occupational exposure (e.g., piggery workers and mosquito surveillance response) [8]; however, many of the JE cases during this outbreak did not fit the current criteria for a vaccination. Thus, immunisation programs would have to target a wider population. Until now, JE immunisation in Australia has been predominantly administered by travel medicine practitioners as subcutaneous or intramuscular injections. JE vaccine uptake among international travellers is low, mainly due to the high cost of JE vaccines (e.g., ~AUD 300 for Imojev and ~AUD 200 for two doses of JEspect) [30].

## 4. Alternative Approaches for a More Economical and Effective Vaccine Delivery

Intradermal (ID) vaccinations using fractional doses have been used for other vaccines such as rabies and BCG to reduce costs and to optimise access during a vaccine shortage [31]. For a few antigens, ID vaccinations using fractional doses may even increase immunogenicity and efficacy whilst being dose-sparing and cost-saving [31]. The use of fractional doses is particularly useful during a vaccine shortage, such as we are currently experiencing with Japanese encephalitis vaccines in Australia [32] and worldwide [33]. An ID vaccination is not a new concept; the first ID trials on the Bacille Calmette–Guérin (BCG) vaccine were in the 1930s, followed by influenza and Yellow Fever in 1940s [34,35]. ID vaccinations using fractional doses have been proven to be safe and effective for several vaccines [36]. Most notably, in recent years, the ID rabies vaccination has been adopted by the Australian Immunisation Handbook (for pre-exposure prophylaxis) [37] and the World Health Organization (for both pre- and post-exposure prophylaxis) [38].

Evidence about the immunogenicity of the ID JE vaccine is currently sparse, but there is potential for its use in the hands of suitably trained providers. ID JE vaccine clinical trials have been conducted using vaccines that are no longer available on the market (e.g., chick embryo-type vaccines), revealing that ID doses were as effective as the standard subcutaneous dose [39]. Currently, our team is conducting a clinical trial (ACTRN12621000024842) on the immunogenicity of Imojev in healthy young adults using a 0.1 mL ID dose (i.e., one fifth of the standard subcutaneous dose). Serological testing using a 50% plaque reduction neutralisation test (PRNT50) revealed that all initial 37 participants seroconverted (i.e., the JE neutralising antibody titre ≥ 10) at one month post-vaccination and the neutralising antibody titres remained stable at two months post-vaccination. Although antibodies persist for >5 years after standard schedules of JE vaccines [40], there is currently no evidence on the long-term persistence of JE antibodies after an ID vaccination. However, this route of vaccine administration has been shown to provide long-term protection for other diseases, including rabies [41] and Yellow Fever [42].

## 5. Conclusions

Australia has experienced a rapid and widespread emergence of the JEV that potentially places large immunologically naïve populations at risk of JE. Currently, there are many knowledge gaps about the long-term effectiveness of the JE vaccination in immunologically naïve populations. Studies on the immunogenicity of ID Imojev and the long-term persistence of antibodies in these populations are important for informing the optimal use of JE vaccines for outbreak control. For mass immunisation programs, two important considerations are vaccine availability and cost. Both challenges might be addressed by using an ID JE vaccination. This will have a regional relevance if outbreaks also occur in neighbouring, less developed countries.

## Figures and Tables

**Figure 1 tropicalmed-07-00085-f001:**
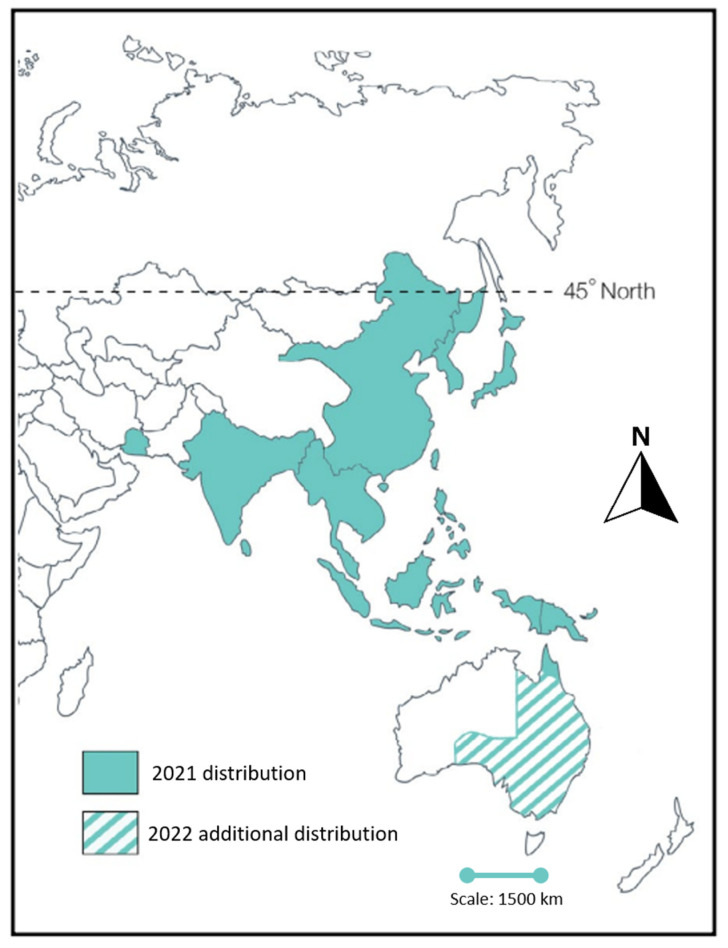
Japanese encephalitis virus is endemic throughout tropical and temperate areas of Asia, up to the 45th parallel. In Australia, local transmission prior to 2021 had been confined to the outer islands of the Torres Strait and Cape York, where it was first identified in 1995. As of May 2022, distribution may have expanded to include large parts of Queensland, New South Wales, Victoria and South Australia. This figure was adapted from the WHO under a creative commons license and is taken, with permissions, from the QIMR Berghofer website.

## Data Availability

The clinical trial is still in progress. Once completed, results and data will be published in a peer-reviewed journal.

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
