# Peer review of "The Emergence of Japanese Encephalitis in Australia and the Implications for a Vaccination Strategy"

_tropicalmed, 2022, doi:10.3390/tropicalmed7060085_

Round 1
Reviewer 1 Report
This is a perspective piece on the emergence of Japanese encephalitis in Australia as a significant public health issue. This is an important topic, and the piece is generally well written and framed. Some suggestions to enhance are provided below.
Abstract says only 5 locally acquired cases prior to 2022, but Introduction lists 6 (including Tiwi Is case).
Reference 5 link not working
Subheading ‘2. Re-emergence of Japanese encephalitis virus’ – consider rewording as is more ‘emergence’ in previously unaffected areas (as indicated by title of piece)
Consider adding a bit more detail for overseas readers on where the various states and territories, Cape York, Torres Strait and Tiwi Islands are, eg could show state/territory boundaries on Fig 1 with arrows to Tiwi/Torres Strait Islands/Cape York.
States “the Australian government has purchased 130,000 JE vaccine doses, but many more may be needed to protect the population at risk if the JE outbreak escalates”. Given cooler weather in southern Australia and reducing mosquito populations and risk, the outbreak is unlikely to ‘escalate’ in the short term. My understanding is however that there are concerns about the virus ‘overwintering’ and reemerging next summer. Hence may be need to consider broader vaccination in meantime, based on risk assessment ie regardless of whether outbreak ‘escalates’. Suggest consider amending to address these nuances.
Statement “For some antigens, ID vaccination using fractional doses could possibly even increase immunogenicity and efficacy while being dose-sparing and cost-saving” sounds a little dubious and is not referenced – suggest either delete or reference appropriately with clarity as to which antigens are referring.
Statement “Most notably, in recent years ID rabies vaccination has been fully adopted by the Australian Immunisation Handbook” is a bit of a stretch – while the Handbook says ID route “may be used by suitably qualified and experienced providers … for pre-exposure vaccination of immunocompetent people” it also says “administration of rabies vaccine for post-exposure prophylaxis by the intradermal route is not recommended.” Suggest reword.
There is a lot of discussion about ID vaccination using fractional doses in context of reducing costs and optimising access during vaccine shortages, however no mention of whether there is a shortage. Suggest consider clarifying eg could reference TGA website in relation to situation in Australia https://apps.tga.gov.au/shortages/search/Details/japanese-encephalitis-virus My understanding is that there are also international manufacturing constraints on supply with long procurement leadtime – eg see UNICEF report at https://www.unicef.org/supply/reports/japanese-encephalitis-vaccine-market-and-supply-update .
Reviewer 2 Report
This is a short communication in which the increase in the number of cases of Japanese encephalitis registered in Australia in 2022 is reported, and aspects of the possible origin of this increase, its geographical distribution and the possibilities of vaccination are discussed.
The main objective of the study is to communicate the significant increase in JE observed in Australia in 2022, and comment on its main implications. It can be useful to alert other areas that could be affected, and eventually travelers who are going to travel to the area.
The article contains single figure that is relevant and correct. I have found no grammatical errors, typos, etc. that needs to be corrected.
The article is well written, and I think it may be of interest for a publication on tropical medicine.
